# GPViT: A High Resolution Non-Hierarchical Vision Transformer with Group Propagation

**Chenhongyi Yang**[*1]  **Jiarui Xu**[*2]  **Shalini De Mello**[3]  **Elliot J. Crowley**[1]  **Xiaolong Wang**[2]

[1]School of Engineering, University of Edinburgh  [2]UC San Diego  [3]NVIDIA

## Abstract

We present the Group Propagation Vision Transformer (GPViT): a novel non-hierarchical (i.e. non-pyramidal) transformer model designed for general visual recognition with high-resolution features. High-resolution features (or tokens) are a natural fit for tasks that involve perceiving fine-grained details such as detection and segmentation, but exchanging global information between these features is expensive in memory and computation because of the way self-attention scales. We provide a highly efficient alternative Group Propagation Block (GP Block) to exchange global information. In each GP Block, features are first grouped together by a fixed number of learnable group tokens; we then perform *Group Propagation* where global information is exchanged between the grouped features; finally, global information in the updated grouped features is returned back to the image features through a transformer decoder. We evaluate GPViT on a variety of visual recognition tasks including image classification, semantic segmentation, object detection, and instance segmentation. Our method achieves significant performance gains over previous works across all tasks, especially on tasks that require high-resolution outputs, for example, our GPViT-L3 outperforms Swin Transformer-B by 2.0 mIoU on ADE20K semantic segmentation with only half as many parameters. Code and pre-trained models are available at https://github.com/ChenhongyiYang/GPViT.

## 1 Introduction

Vision Transformer (ViT) architectures have achieved excellent results in general visual recognition tasks, outperforming ConvNets in many instances. In the original ViT architecture, image patches are passed through transformer encoder layers, each containing self-attention and MLP blocks. The spatial resolution of the image patches is constant throughout the network. Self-attention allows for information to be exchanged between patches across the whole image i.e. globally, however it is computationally expensive and does not place an emphasis on local information exchange between nearby patches, as a convolution would. Recent work has sought to build convolutional properties back into

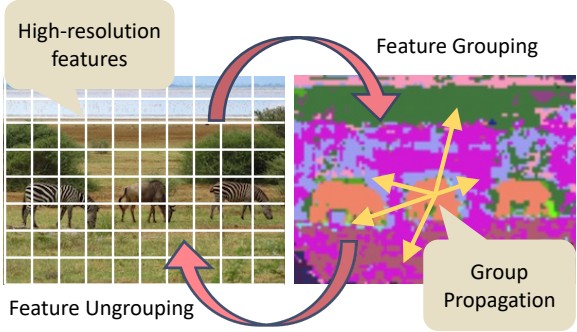

Figure 1: An illustration of our GP Block. It groups image features into a fixed-size feature set. Then, global information is efficiently propagated between the grouped features. Finally, the grouped features are queried by the image features to transfer this global information into them.

vision transformers (Liu et al., 2021; Wu et al., 2021; Wang et al., 2021) through a hierarchical (pyramidal) architecture. This design reduces computational cost, and improves ViT performance on tasks such as detection and segmentation.

Is this design necessary for structured prediction? It incorporates additional inductive biases e.g. the assumption that nearby image tokens contains similar information, which contrasts with the

---

[*]Equal Contribution

motivation for ViTs in the first place. A recent study (Li et al., 2022a) demonstrates that a plain non-hierarchical ViT, a model that maintains the same feature resolution in all layers (non-pyramidal), can achieve comparable performance on object detection and segmentation tasks to a hierarchical counterpart. How do we go one step further and *surpass* this? One path would be to increase feature resolution (i.e. the number of image tokens). A plain ViT with more tokens would maintain high-resolution features throughout the network as there is no downsampling. This would facilitate fine-grained, detailed outputs ideal for tasks such as object detection and segmentation. It also simplifies the design for downstream applications, removing the need to find a way to combine different scales of features in a hierarchical ViT. However, this brings new challenges in terms of computation. Self-attention has quadratic complexity in the number of image tokens. Doubling feature resolution (i.e. quadrupling the number of tokens) would lead to a $16\times$ increase in compute. How do we maintain global information exchange between image tokens without this huge increase in computational cost?

In this paper, we propose the **Group Propagation Vision Transformer (GPViT)**: a non-hierarchical ViT which uses high resolution features throughout, and allows for efficient global information exchange between image tokens. We design a novel Group Propagation Block (GP Block) for use in plain ViTs. Figure 1 provides a high-level illustration of how this block works. In detail, we use learnable group tokens and the cross-attention operation to group a large number of high-resolution image features into a fixed number of grouped features. Intuitively, we can view each group as a cluster of patches representing the same semantic concept. We then use an MLPMixer (Tolstikhin et al., 2021) module to update the grouped features and propagate global information among them. This process allows information exchange at a low computational cost, as the number of groups is much smaller than the number of image tokens. Finally, we ungroup the grouped features using another cross-attention operation where the updated grouped features act as key and value pairs, and are queried by the image token features. This updates the high resolution image token features with the group-propagated information. The GP Block only has a linear complexity in the number of image tokens, which allows it to scale better than ordinary self-attention. This block is the foundation of our simple non-hierarchical vision transformer architecture for general visual recognition.

We conduct experiments on multiple visual recognition tasks including image classification, object detection, instance segmentation, and semantic segmentation. We show significant improvements over previous approaches, including hierarchical vision transformers, under the same model size in all tasks. The performance gain is especially large for object detection and segmentation. For example, in Figure 2, we show GPViT's advantage over the non-hierarchical DeiT (Touvron et al., 2021a) and hierarchical Swin Transformer (Liu et al., 2021) on those recognition tasks. In addition, our smallest model GPViT-L1 can outperform the Swin Transformer-B (Liu et al., 2021) by 2.6 $AP^{bb}$ and 1.4$^{mk}$ in COCO Mask R-CNN (He et al., 2017) object detection and instance segmentation with only 30% as many parameters, and GPViT-L2 outperforms Swin Transformer-B by 0.5 mIoU on UperNet (Xiao et al., 2018) ADE20K semantic segmentation also with only 40% as many parameters.

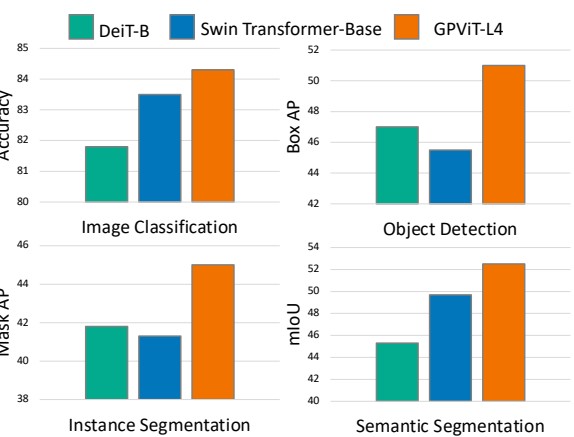

Figure 2: A comparison on four visual recognition tasks between GPViT and the non-hierarchical DeiT (Touvron et al., 2021a) and the hierarchical Swin Transformer (Liu et al., 2021).

## 2 RELATED WORK

**Vision Transformers.** Vision Transformers have shown great success in visual recognition. They have fewer inductive biases, e.g. translation invariance, scale-invariance, and feature locality (Xu et al., 2021b) than ConvNets and can better capture long-range relationships between image pixels. In the original ViT architecture (Dosovitskiy et al., 2021; Touvron et al., 2021a), images are split into patches and are transformed into tokens that are passed through the encoder of a transformer (Vaswani et al.,

2017). Based on this framework, LeViT (Graham et al., 2021) achieves a significant performance improvement over ViT by combining convolutional and transformer encoder layers. An important development in ViT architectures is the incorporation of a hierarchical feature pyramid structure, as typically seen in ConvNets (Wang et al., 2021; Liu et al., 2021; Xu et al., 2021a; Wu et al., 2021; Fan et al., 2021). For example, Liu et al. (2021) propose a shifted windowing scheme to efficiently propagate feature information in the hierarchical ViT. Such a pyramid architecture provides multi-scale features for a wide range of visual recognition tasks. Following this line of research, recent work has studied the use of hierarchical features in ViTs (Ren et al., 2022b; Guo et al., 2022; Li et al., 2022b; Dong et al., 2022; Hatamizadeh et al., 2022; Chen et al., 2022a; d'Ascoli et al., 2021; Lee et al., 2022). For example, Ren et al. (2022b) introduce using multi-resolution features as attention keys and values to make the model learn better multi-scale information. While this is encouraging, it introduces extra complexity in the downstream model's design on how to utilize the multi-scale features effectively. Recently, Li et al. (2022a) revisited the plain non-hierarchical ViT for visual recognition; using such a model simplifies the use of features and better decouples the pre-training and downstream stages of model design. Our work extends on this as we examine how to efficiently increase the feature resolution in a non-hierarchical ViT.

**Attention Mechanisms in ViTs.** A bottleneck when using high resolution features in ViTs is the quadratic complexity in the computation of the self-attention layer. To tackle this challenge, several local attention mechanisms have been proposed (Liu et al., 2021; Huang et al., 2019; Dong et al., 2022; Xu et al., 2021a; Zhang et al., 2022; Han et al., 2021) to allow each image token to attend to local region instead of the whole image. However, using only local attention hinders a model's ability to exchange information globally. To counter this problem, RegionViT (Chen et al., 2022a) and GCViT (Hatamizadeh et al., 2022) first down-sample their feature maps and exchange global information between the down-sampled features, before using self-attention to transfer information between the original image features and the down-sampled features. This is similar in spirit to our GP Block. However, unlike RegionViT and GCViT, in a GP Block the grouped features are not constrained to a particular rectangular region, but can correspond to any shape or even entirely disconnected image parts. There is recent work using transformer decoder layers with cross-attention between visual tokens and learnable tokens (Carion et al., 2020; Cheng et al., 2022; Jaegle et al., 2022; Hudson & Zitnick, 2021), however, there are three fundamental differences between these and ours: (i) Each of our GP blocks operates as an 'encoder-decoder' architecture with two rounds of cross-attention between visual tokens and group tokens: the first round groups the visual tokens for group propagation, and the second round ungroups the updated groups back into visual tokens; (ii) The underlying functionality is different: GP blocks facilitate more efficient global information propagation throughout the ViT, while previous work applies the decoder to obtain the final results for inference (e.g bounding boxes, or masks in Carion et al. (2020); Cheng et al. (2022)); (iii) The GP block is a general module that can be insert into any layer of the ViT, while previous work utilizes the decoder only in the end of the network.

**High-Resolution Visual Recognition.** Previous work (Wang et al., 2020; Cheng et al., 2020) has shown that high-resolution images and features are beneficial to visual recognition tasks, especially to those requiring the perception of fine-grained image details, for example, semantic segmentation (Wang et al., 2020), pose-estimation (Sun et al., 2019), and small object detection (Yang et al., 2022). For example, HRNet (Wang et al., 2020) introduces a high-resolution ConvNet backbone. It maintains a high-resolution branch and exchanges information between different resolutions of features with interpolation and strided convolutions. Inspired by this work, HRFormer (Yuan et al., 2021) and HRViT (Gu et al., 2022) replace the convolutions in HRNet with self-attention blocks. GPViT is even simpler: it maintains single-scale and high-resolution feature maps without requiring any cross-resolution information to be maintained.

**Object-Centric Representation.** Our idea of performing information propagation among grouped regions is related to object-centric representation learning (Wang & Gupta, 2018; Kipf & Welling, 2017; Watters et al., 2017; Qi et al., 2021; Locatello et al., 2020; Kipf et al., 2022; Elsayed et al., 2022; Xu et al., 2022). For example, Locatello et al. (2020) proposes slot-attention, which allows automatic discovery of object segments via a self-supervised reconstruction objective. Instead of using reconstruction, Xu et al. (2022) utilizes language as an alternative signal for object segmentation discovery and shows it can be directly transferred to semantic segmentation in a zero-shot manner. All the above work extract object-centric features for downstream applications, while our work inserts this object-centric information propagation mechanism as a building block inside ViTs to compute

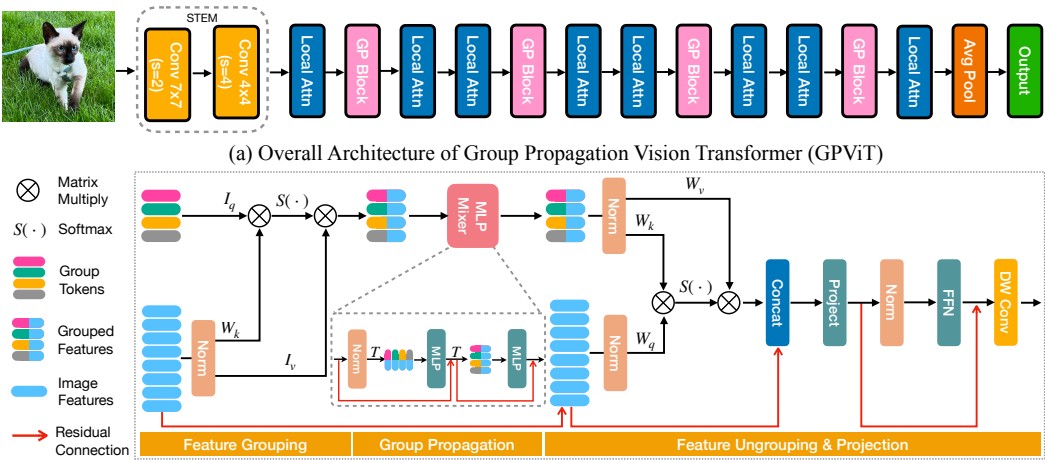

(a) Overall Architecture of Group Propagation Vision Transformer (GPViT)

(b) Architecture of Group Propagation Block (GP Block)

Figure 3: (a). GPViT architecture: The input image is first fed to a convolutional stem that downsamples by a factor of 8. Each pixel of this downsampled image is treated as a high resolution image token or *feature*, and positional embeddings are added (not shown in figure). These features are fed into our transformer, which consists of 12 encoder layers. 8 of these use local attention, and 4 use our proposed GP Block to propagate global information between image features. (b). GP Block: Image features are grouped using a fixed number of learnable group tokens. An MLPMixer module is then used to exchange global information and update the grouped features. Next, the grouped features are queried by, and concatenated with the image features to transfer global information to every image feature. Finally the updated image features are transformed by a feed-forward network to produce the output.

high-resolution representations more efficiently and improve high-resolution features. In this respect, our work is related to Li & Gupta (2018) where the graph convolution operations are inserted into ConvNets for better spatial reasoning.

## 3 METHOD

We present the overall architecture of our Group Propagation Vision Transformer (GPViT) in Figure 3 (a). GPViT is designed for general high-resolution visual recognition. For stable training, we first feed the input image into a down-sampling convolutional stem to generate image features (also known as image tokens), as in Dosovitskiy et al. (2021); Liu et al. (2021). In GPViT we downsample by a factor of 8 by default. The features are therefore higher resolution than in the original ViT where the factor is 16. Unlike most recently proposed methods (Liu et al., 2021; Li et al., 2022b) that adopt a pyramid structure to generate features in multiple resolutions, we keep the features at a high resolution without any down-sampling.

After combining the initial image features with positional embeddings (Vaswani et al., 2017), we feed them into the core GPViT architecture. We replace the original self-attention block in ViT with local attention to avoid the quadratic complexity of self-attention. However, stacking local attention blocks alone does not allow for long-range information exchange between patches and therefore is harmful to performance. To counter this problem, we propose the Group Propagation Block (GP Block)—which we describe in full in Section 3.1—to efficiently propagate global information across the whole image. In our implementation, we use a mixture of GP Blocks and local attention layers to form our GPViT and keep the overall depth unchanged. Lastly, we average the final features to get the model's output.

### 3.1 GROUP PROPAGATION BLOCK

Our key technical contribution is the GP block, which efficiently exchanges global information between each image patch with a linear complexity. We visualize the structure of the GP block in Figure 3 (b). It has a bottleneck structure and comprises of three stages, namely, *Feature Grouping*, *Group Propagation*, and *Feature Ungrouping*. In the first stage the image features are grouped, then in the second stage global information is propagated between the grouped features, and in the last stage, this global information is transferred back to the image features.

**Feature Grouping.** The input to a GP Block is a matrix of image features $X \in \mathbb{R}^{N \times C}$ (The blue tokens in Figure 3 (b)) where $N$ is the total number of image features (or image tokens) and $C$ is the dimensionality of each feature vector. We use $M$ learnable group tokens stored in a matrix $G \in \mathbb{R}^{M \times C}$ (the multi-colored tokens in Figure 3 (b)) where the group number $M$ is a model hyper-parameter. Grouping is performed using a simplified multi-head attention operation (Vaswani et al., 2017), which gives us grouped features $Y \in \mathbb{R}^{M \times C}$ (the half-and-half tokens in Figure 3 (b)):

$$\text{Attention}(Q, K, V) = \text{Softmax}(\frac{QK^T}{\sqrt{d}})V, \tag{1}$$

$$Y = \text{Concat}_{\{h\}}\big(\text{Attention}(W_h^Q G_h, W_h^K X_h, W_h^V X_h)\big), \tag{2}$$

where $d$ is the channel number, $h$ is the head index, and $W_h^{\{Q,K,V\}}$ are projection matrices for the query, key, and values, respectively in the attention operation. We remove the feature projection layers after the concatenation operation and set $W_h^Q$ and $W_h^V$ to be identity matrix. Therefore, the grouped features are simply the weighted sum of image features at each head where the weights are computed by the attention operation.

**Group Propagation.** After acquiring the grouped features, we can update and propagate global information between them. We use an MLPMixer (Tolstikhin et al., 2021) (Equation 3; the red box in Figure 3 (b)) to achieve this, as MLPMixer provides a good trade-off between model parameters, FLOPs, and model accuracy. MLPMixer requires a fixed-sized input, which is compatible with our fixed number of groups. Specifically, our MLPMixer contains two consecutive MLPs. Recall that $Y \in \mathbb{R}^{M \times C}$ contains the grouped features from the first *Feature Grouping* stage. We can update these features to $\tilde{Y} \in \mathbb{R}^{M \times C}$ with the MLPMixer by computing:

$$Y' = Y + \text{MLP}_1(\text{LayerNorm}(Y)^T))^T, \tag{3}$$

$$\tilde{Y} = Y' + \text{MLP}_2(\text{LayerNorm}(Y'))), \tag{4}$$

where the first MLP is used for mixing information between each group, and the second is used to mix channel-wise information.

**Feature Ungrouping.** After updating the grouped features, we can return global information to the image features through a *Feature Ungrouping* process. Specifically, the features are ungrouped using a transformer decoder layer where grouped features are queried by the image features.

$$U = \text{Concat}_{\{h\}}\big(\text{Attention}(\tilde{W}_h^Q X_h, \tilde{W}_h^K \tilde{Y}_h, \tilde{W}_h^V \tilde{Y}_h)\big), \tag{5}$$

$$Z' = W_{proj} * \text{Concat}(U, X), \quad Z'' = Z' + \text{FFN}(Z'), \quad Z = \text{DWConv}(Z''), \tag{6}$$

where $\tilde{W}_h^{\{Q,K,V\}}$ are the projection matrices in the attention operation, $W_{proj}$ is a linear matrix that projects concatenated features $Z'$ to the same dimension as image features $X$, FFN is a feed-forward network, and DWConv is a depth-wise convolution layer. We modify the original transformer decoder layer by replacing the first residual connection with a concatenation operation (Equation 5; the blue box in Figure 3 (b)), and move the feature projection layer after this to transform the feature to the original dimension. We find this modification benefits the downstream tasks in different sizes of models. We take inspiration from Ren et al. (2022b) and add a depth-wise convolution at the end of the GP Block to improve the locality property of the features (Equation 6; the yellow box in Figure 3 (b)). Finally, a GP Block outputs $Z$ as its final output.

Table 1: GPViT architecture variants.

| Model | Channels | Param (M) | FLOPs (G) |
|---|---|---|---|
| GPViT-L1 | 216 | 9.3 | 5.8 |
| GPViT-L2 | 348 | 23.6 | 15.0 |
| GPViT-L3 | 432 | 36.2 | 22.9 |
| GPViT-L4 | 624 | 75.4 | 48.2 |

## 3.2 ARCHITECTURE VARIANTS OF GPVIT

In this paper we study four variants of the proposed GPViT. We present their architectural details in Table 1. These four variants largely differ in the number of feature channels used (i.e. the model width). We use the recently proposed LePE attention (Dong et al., 2022) as local attention by default. The FLOPs are counted using 224×224 inputs. Please refer to Section B.1 in our Appendix for detailed architectural hyper-parameters and training recipes for these variants.

## 3.3 COMPUTATIONAL COSTS OF HIERARCHICAL AND NON-HIERARCHICAL ViTs.

We visualize both the non-hierarchical and hierarchical ViT in Figure 4 (a), where the non-hierarchical ViT simply stacks attention blocks and the hierarchical ViT divides the network into several stages and down-samples the feature map at each stage. Naturally, with the same resolution input, the non-hierarchical ViT will have a higher computation cost. The cost is divided into two parts as shown in Figure 4 (b): the self-attention module and the FFN module. Our GPViT largely reduces the computation of information propagation by using our GP Block instead of self-attention. However, the cost of the FFN stays high for high resolution image features. Therefore we will expect higher FLOPs from GPViT compared to a hierarchical ViT given similar model parameters. However, we believe non-hierarchical ViTs are still a direction worthy of exploration given their simplicity in extracting high-resolution features and the removal of the need to study the design of efficient downstream models that utilize multi-scale features as required for a hierarchical ViT. This helps to maintain the independence of the model's pre-training and fine-tuning designs (Li et al., 2022a). In our experiments, we show that our GPViT can achieve better detection and segmentation performance compared to state-of-the-art hierarchical ViTs with similar FLOP counts.

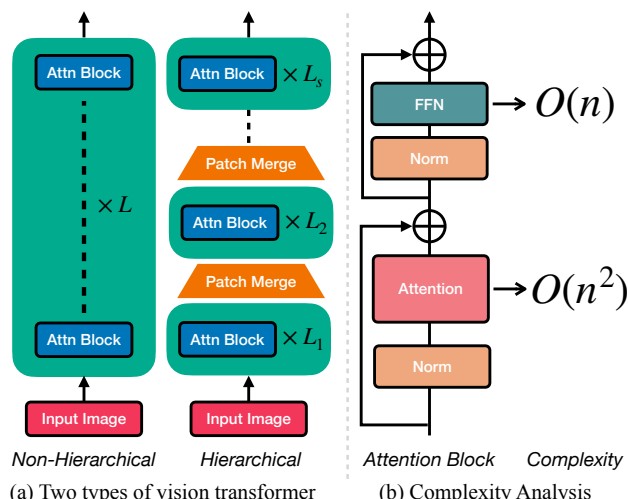

(a) Two types of vision transformer  (b) Complexity Analysis

Figure 4: (a). Structure comparison between non-hierarchical and hierarchical ViTs. (b). The computation cost of an attention block.

## 4 EXPERIMENTS

### 4.1 IMAGENET-1K CLASSIFICATION

**Setting:** To ensure a fair comparison with previous work, we largely follow the training recipe of Swin Transformer (Liu et al., 2021). We build models using the MMClassification (Contributors, 2020a) toolkit. The models are trained for 300 epochs with a batch size of 2048 using the AdamW optimizer with a weight decay of 0.05 and a peak learning rate of 0.002. A cosine learning rate schedule is used to gradually decrease the learning rate. We use the data augmentations from Liu et al. (2021); these include Mixup (Zhang et al., 2017), Cutmix (Yun et al., 2019), Random erasing (Zhong et al., 2020) and Rand augment (Cubuk et al., 2020).

We compare GPViT with hierarchical and non-hierarchical vision transformers on the ImageNet-1K classification task and report the results in Table 2. As shown in the table, because of our high-resolution design and effective global information propagation via the grouping mechanism, our GPViT outperforms outperforms the non-hierarchical baseline DeiT (Touvron et al., 2021a). In addition, GPViT also outperforms Swin Transformer (Liu et al., 2021) and two recently proposed hierarchical counterparts RegionViT (Chen et al., 2022a) and DWViT (Ren et al., 2022a). This result showcases the potential of non-hierarchical vision transformers and suggests that the hierarchical design inherited from the ConvNet era is not necessary for obtaining a high-performing visual recognition model. This corroborates the work of Li et al. (2022a). That said, we do note that the FLOPs of our models are higher than most alternatives for a

Table 2: Comparison between GPViT and the recent proposed models on ImageNet-1K.

| Model | Params (M) | FLOPs (G) | Top-1 Acc |
|---|---|---|---|
| **Hierarchical** | | | |
| Swin-T (Liu et al., 2021) | 29.0 | 4.5 | 81.3 |
| Swin-B (Liu et al., 2021) | 88 | 15.4 | 83.5 |
| RegionViT-S (Chen et al., 2022a) | 30.6 | 5.3 | 82.6 |
| RegionViT-B (Chen et al., 2022a) | 72.7 | 13.0 | 83.2 |
| DW-T (Ren et al., 2022a) | 30.0 | 5.2 | 82.0 |
| DW-B (Ren et al., 2022a) | 91.0 | 17.0 | 83.8 |
| **Non-hierarchical** | | | |
| DeiT-S (Touvron et al., 2021a) | 22.1 | 4.6 | 79.9 |
| DeiT-B (Touvron et al., 2021a) | 86 | 16.8 | 81.8 |
| ConViT-S (d'Ascoli et al., 2021) | 27.0 | 5.4 | 81.3 |
| ConViT-B (d'Ascoli et al., 2021) | 86.0 | 17.0 | 82.4 |
| GPViT-L1 | 9.3 | 5.8 | 80.5 |
| GPViT-L2 | 23.8 | 15.0 | 83.4 |
| GPViT-L3 | 36.2 | 22.9 | 84.1 |
| GPViT-L4 | 75.4 | 48.2 | 84.3 |

Table 3: Mask R-CNN object detection and instance segmentation on MS COCO *mini-val* using 1× and 3× (or longer) + MS schedule.

| Backbone | Params (M) | FLOPs (G) | 1× $AP^{bb}$ | 1× $AP^{mk}$ | 3× or more $AP^{bb}$ | 3× or more $AP^{mk}$ |
|---|---|---|---|---|---|---|
| **Hierarchical** | | | | | | |
| RegionViT-B (Chen et al., 2022a) | 92 | 287 | 44.2 | 40.8 | 47.6 | 43.4 |
| Swin-B (Liu et al., 2021) | 107 | 496 | 45.5 | 41.3 | - | - |
| DaViT-Base (Ding et al., 2022) | 107 | 491 | 48.2 | 43.3 | 49.9 | 44.6 |
| DW-B (Ren et al., 2022a) | 111 | 505 | - | - | 49.2 | 44.0 |
| CSwin-B (Dong et al., 2022) | 97 | 526 | 48.7 | 43.9 | 50.8 | 44.9 |
| MPViT-B (Lee et al., 2022) | 95 | 503 | - | - | 49.5 | 44.5 |
| **Non-hierarchical** | | | | | | |
| ViT-Adapter-B (Chen et al., 2022b) | 102 | - | 47.0 | 41.8 | 49.6 | 43.6 |
| ViTDet-SUP* (Li et al., 2021a) | 111 | 800 | - | - | 47.6 | 42.4 |
| ViTDet-MAE* (Li et al., 2021a) | 111 | 800 | - | - | 51.6 | 45.2 |
| GPViT-L1 | 33 | 457 | 48.1 | 42.7 | 50.2 | 44.3 |
| GPViT-L2 | 50 | 690 | 49.9 | 43.9 | 51.4 | 45.1 |
| GPViT-L3 | 64 | 884 | 50.4 | 44.4 | 51.6 | 45.2 |
| GPViT-L4 | 109 | 1489 | 51.0 | 45.0 | 52.1 | 45.7 |

*: ViTDet (Li et al., 2021a) models were trained for 100 epochs with advanced regularisation techniques.

similar parameter count. However, for a similar FLOP count we observe that GPViT can achieve a comparable top-1 accuracy, but with many fewer parameters than the alternatives. For example, GPViT-L2 (15.0 G) has similar FLOPs to the Swin Transformer-B (15.4 G) and ShiftViT-B (15.6 G), but it achieves a similar accuracy with significantly fewer parameters (23.8 M v.s. 88 M and 89 M).

## 4.2 COCO OBJECT DETECTION AND INSTANCE SEGMENTATION

**Setting:** We follow Chen et al. (2022b) to use Mask R-CNN and RetinaNet models for the COCO object detection and instance segmentation tasks. We use ViTAdapter (Chen et al., 2022b) to generate multi-scale features as FPN inputs and evaluate the model for both 1× and 3× training schedules.

**Results:** We compare GPViT to state-of-the-art backbones, all pre-trained on ImageNet-1K. We report the results in Table 3 and Table 4. For competing methods we report the performance of their largest-sized models. For both detectors our GPViT is able to surpass the other backbones by a large margin for a similar parameter count. With Mask R-CNN (Table 3), our smallest GPViT-L1 surpasses its Swin Transformer-B (Liu et al., 2021) counterpart by 2.6 $AP^{bb}$ and 1.4 $AP^{mk}$ for the 1× training schedule with fewer FLOPs and only 30% as many parameters. When comparing with models that are also equipped with ViTAdapter (Chen et al., 2022b), we observe that GPViT achieves a better

Table 4: RetinaNet object detection on MS COCO *mini-val* with 1× and 3× +MS schedule.

| Backbone | Params (M) | FLOPs (G) | 1× $AP^{bb}$ | 3× $AP^{bb}$ |
|---|---|---|---|---|
| **Hierarchical** | | | | |
| PVT-L (Wang et al., 2021) | 71 | 345 | 42.6 | - |
| PVTv2-B5 (Wang et al., 2022b) | 91 | 335 | 46.1 | - |
| Swin-B (Liu et al., 2021) | 98 | 477 | 44.7 | - |
| RegionViT-B (Chen et al., 2022a) | 83 | 308 | 43.3 | 46.1 |
| MPViT-B (Lee et al., 2022) | 95 | 503 | - | 48.3 |
| DaViT-Base (Ding et al., 2022) | 103 | 471 | 46.7 | 48.7 |
| **Non-hierarchical** | | | | |
| GPViT-L1 | 21 | 317 | 45.8 | 48.1 |
| GPViT-L2 | 37 | 542 | 48.0 | 49.0 |
| GPViT-L3 | 52 | 731 | 48.3 | 49.4 |
| GPViT-L4 | 96 | 1319 | 48.7 | 49.8 |

AP with fewer parameters, e.g. our smallest GPViT-L1 outperforms ViT-Adapter-B in both training schedules. These results showcase GPViT's effectiveness at extracting good regional features for object detection and instance segmentation. A similar conclusion can be drawn from the single-stage RetinaNet detector; with RetinaNet (Table 4), GPViT-L1 has FLOPs similar to the recently proposed RegionViT-B (Chen et al., 2022a), but it outperforms RegionViT-B by 2.5 and 2.0 $AP^{bb}$ in both 1× and 3× schedules with only 25% as many parameters. In Table 3, we also compare our Mask R-CNN with the recently proposed ViTDet (Li et al., 2021a) that also uses a non-hierarchical ViT as the backbone network. Here we continue to use the standard 3× (36 epochs) training recipe for GPViT. The results show that under similar FLOPs, even if ViTDet is equipped with more parameters (111M), advanced masked-auto-encoder (MAE) pre-training (He et al., 2022), a longer training schedule (100 epochs), and heavy regularizations like large-scale jittering (Ghiasi et al., 2021), our model can still achieve a comparable performance, which further validates the effectiveness of GPViT.

Table 5: Comparison between GPViT and other vision transformers on ADE20K semantic segmentation task.

| UperNet | | | | SegFormer | | | |
|---|---|---|---|---|---|---|---|
| | Params | FLOPs | | | Params | FLOPs | |
| Backbone | (M) | (G) | mIoU | Backbone | (M) | (G) | mIoU |
| Hierarchical | | | | Hierarchical | | | |
| Swin-B (Liu et al., 2021) | 121 | 1188 | 49.7 | MiT-B2 (Xie et al., 2021) | 24 | 64 | 46.5 |
| DAT-T (Xia et al., 2022) | 121 | 1212 | 50.5 | MiT-B4 (Xie et al., 2021) | 61 | 96 | 50.4 |
| DaViT-Base (Ding et al., 2022) | 121 | 1175 | 49.4 | Stunned-S (Ren et al., 2022b) | 25 | 78 | 48.3 |
| MPViT-B (Lee et al., 2022) | 105 | 1186 | 50.3 | CSwin-S(Dong et al., 2022) | 37 | 78 | 49.9 |
| DW-B (Ren et al., 2022a) | 125 | 1200 | 48.7 | HRViT-b3(Gu et al., 2022) | 29 | 68 | 50.2 |
| Non-hierarchical | | | | HILA+MiT-B1 (Leung et al., 2022) | 22 | 31 | 45 |
| | | | | HILA+MiT-B2 (Leung et al., 2022) | 31 | 76 | 46 |
| Deit-B (Touvron et al., 2021a) | 120 | 786 | 45.3 | Non-hierarchical | | | |
| ViT-Adapter-B (Chen et al., 2022b) | 134 | - | 48.1 | GPViT-L1 | 9 | 34 | 46.9 |
| GPViT-L1 | 37 | 568 | 49.1 | GPViT-L2 | 24 | 83 | 49.2 |
| GPViT-L2 | 53 | 775 | 50.2 | GPViT-L3 | 36 | 123 | 50.8 |
| GPViT-L3 | 66 | 944 | 51.7 | GPViT-L4 | 76 | 249 | 51.3 |
| GPViT-L4 | 107 | 1469 | 52.5 | | | | |

## 4.3 ADE20K SEMANTIC SEGMENTATION

**Setting:** We follow previous work (Liu et al., 2021) and use UperNet (Xiao et al., 2018) as the segmentation network. We also report performance when using the recently proposed SegFormer (Xie et al., 2021) model. For both models, we train for 160k iterations.

**Results:** We summarise the segmentation performance of GPViT and other state-of-the-art backbone networks in Table 5. For UperNet, we report results with the largest available model size for the competing methods to show how far we can go in the segmentation task. Thanks to its high-resolution design, GPViT outperforms all competing methods in mIoU with fewer FLOPs and fewer parameters. For example, GPViT-L1 only has 37M parameters but it can achieve comparable mIoU to methods with only half the number of FLOPs. This result tells us that for tasks requiring the perception of fine-grained details, scaling-up feature resolution is a better strategy than scaling up model size. GPViT also excels when used with SegFormer. Specifically, GPViT achieves better mIoU than recently proposed vision transformers with similar parameter counts, including HRViT (Gu et al., 2022) that was specifically designed for semantic segmentation. We attribute these promising results to GPViT's high-resolution design and its effective encapsulation of global information.

## 4.4 ABLATION STUDIES

**Setting:** We conduct ablation studies using two types of local attention: the simple window attention (Liu et al., 2021) and the more advanced LePE attention (Dong et al., 2022), which we used in previous experiments. We use the L1 level models (Param < 10 M) for all experiments. All models are pre-trained on ImageNet classification task for 300 epochs using the same setting as in Section 4.1. We report both ImageNet Top-1 accuracy and ADE20K SegFormer mIOU. Please refer to our appendix for more ablation experiments.

**Building GPViT step by step.** Here we show how we build GPViT step by step and present the results in Table 6. We start building our GPViT from a low-resolution vanilla DeiT with patch sizes of 16 and embedding channels of 216 (same as GPViT-Tiny). It achieves 77.4 top-1 accuracy on ImageNet and 42.2 mIoU on ADE20K. Then we increase the resolution by shrinking the patch size to 8. The FLOPs of the ImageNet and ADE20K models increase by 4.4× and 7.0× respectively. ImageNet accuracy increases to 79.2 but training this model for segmentation proves

Table 6: Ablation studies on feature resolution and GP Block using two different local attentions.

| Setting | FLOPs (G) | Top-1 Acc | FLOPs (G) | mIoU |
|---|---|---|---|---|
| ViT-D216-P16 | 1.8 | 77.4 | 16 | 42.2 |
| ViT-D216-P8 | 8.8 | 79.2 | 113 | - |
| + Win attn | 5.8 | 78.1 | 34 | 41.7 |
| + GP Block | 5.8 | 79.8 | 34 | 45.5 |
| + LePE attn | 5.8 | 79.5 | 34 | 46.2 |
| + GP Block | 5.8 | 80.5 | 34 | 46.9 |

to be unstable. We see that enlarging the feature resolution using global self-attention leads to the number of FLOPs exploding and makes convergence difficult. We now replace self-attention with window attention (Liu et al., 2021) and the more advanced LePE attention (Dong et al., 2022). For both local attention mechanisms, the FLOPs of the ImageNet and ADE20K models drop to 5.8G and 34G respectively. We then incorporate GP Blocks, and observe that the accuracy and mIoU improve for both types of local attention and FLOPs remain unchanged. These results showcase the effectiveness of using high-resolution features as well as the importance of our combination of local attention blocks and GP Blocks to maintain a reasonable computation cost.

**Global information exchange.** Here, we compare our GP Block with other blocks that can exchange global information between image features. The competing blocks include the global attention block, the convolution propagation block (Li et al., 2022a), and the shifting window-attention block (Liu et al., 2021) designed for window attention. We follow ViT-Det (Li et al., 2022a) to build the convolution propagation block that stacks two 3×3 convolution layers with a residual connection. We use the original version of the shifting window-attention block as in Liu et al. (2021). The resulting models are acquired by putting competing

Table 7: Ablation study on different global information propagation methods.

| Attention | Global Info | FLOPs (G) | Top-1 Acc | FLOPs (G) | mIoU |
|---|---|---|---|---|---|
| Window | None | 5.8 | 78.2 | 34 | 41.7 |
| | Conv | 6.6 | 75.8 | 38 | 39.5 |
| | Global attn | 6.8 | 78.8 | 62 | 44.0 |
| | Win shift | 5.8 | 78.1 | 34 | 40.7 |
| | GP Block | 5.8 | 79.8 | 34 | 45.5 |
| LePE | None | 5.8 | 79.5 | 34 | 46.2 |
| | Conv | 6.6 | 77.7 | 38 | 45.5 |
| | Global attn | 6.8 | 80.4 | 62 | 46.7 |
| | GP Block | 5.8 | 80.5 | 34 | 46.9 |

blocks in the same place as as our GP Block. We report the results in Table 7. We observe that simply replacing the local attention layers with convolution layers causes severe performance drops for both types of local attention.We also observe that replacing local attention with global attention can improve performance for a very large increase in FLOPs. For window attention, we found that using the shifting window strategy slightly hurts the performance. We postulate that this is caused by a deficit of shifting window layers; half of Swin Transformer layers are shifting window layers, but we only use four here. For both types of local attention, GP Block achieves the best performance on ImageNet and ADE20K. These results show GP Block's effectiveness in propagating global information.

**Number of group tokens.** Here we study how the different combinations of the number of groups tokens in GP Blocks affect the overall model performance. We report the results in Table 8. We find using a large number of group tokens across the whole network can give us higher accuracy on ImageNet but at additional computational cost. However, using too few group e.g. 16 tokens will harm performance. In GPViT we choose to progressively decrease the

Table 8: Ablation study on the group tokens number combinations.

| Combination | FLOPs (G) | Top-1 Acc |
|---|---|---|
| {16, 16, 16, 16} | 5.7 | 79.9 |
| {32, 32, 32, 32} | 5.8 | 80.3 |
| {64, 64, 64, 64} | 6.0 | 80.7 |
| {16, 32, 32, 64} | 5.8 | 80.0 |
| {64, 32, 32, 16} | 5.8 | 80.5 |

number of group tokens from 64 to 16. This strategy gives us a good trade-off between accuracy and computational cost.

**Grouped features propagation.** In Table 9 we compare different methods for global information propagation. The results show that even when we add nothing to explicitly propagate global information the model can still achieve a good performance (79.8% accuracy on ImageNet). The reason is that in this case the image features are still grouped and ungrouped so the

Table 9: Ablation study on the propagation approach of grouped features.

| Method | FLOPs (G) | Top-1 Acc |
|---|---|---|
| None | 5.7 | 79.8 |
| SelfAttn | 6.2 | 80.7 |
| MLPMixer | 5.8 | 80.5 |

global information can still be exchanged in these two operations. We also find that self-attention achieve a slightly better accuracy than MLPMixer (80.7 v.s. 80.5), but is more expensive. In GPViT we use MLPMixer for propagating global information.

## 5 CONCLUSION

In this paper, we have presented the Group Propagation Vision Transformer (GPViT): a non-hierarchical vision transformer designed for high-resolution visual recognition. The core of GPViT is the GP Block, which was proposed to efficiently exchange global information among high-resolution features. The GP Block first forms grouped features and thxen updates them through *Group Propagation*. Finally, these updated group features are queried back to the image features. We have shown that GPViT can achieve better performance than previous work on ImageNet classification, COCO object detection and instance segmentation, and ADE20K semantic segmentation.

## 6 ACKNOWLEDGEMENT

Prof. Xiaolong Wang's group was supported, in part, by gifts from Qualcomm and Amazon Research Award. Chenhongyi Yang was supported by a PhD studentship provided by the School of Engineering, University of Edinburgh.

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

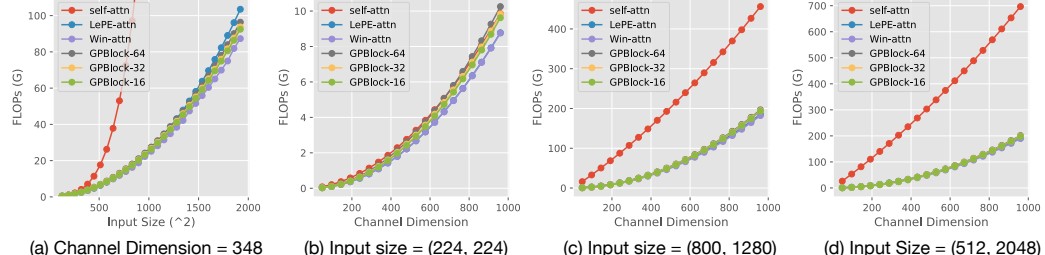

Figure 5: Layer-wise FLOPs comparison between different attention blocks on high-resolution features. "GPBlock-$N$" denotes GP Block with $N$ group tokens. (a) FLOPs v.s. Input size when using feature channel dimension of 348 (same as GPViT-s1); (b-d): FLOPs v.s. feature channel dimension on three typical input sizes using by ImageNet classification, COCO object detection and instance segmentation, and ADE20K semantic segmentation. Our GP Block can effectively gather and propagate global information like a self-attention block while having similar or even fewer FLOPs with local attention blocks when the model or input scale-up.

Table 10: Inference speed comparison on three typical input sizes: $224{\times}224$ for ImageNet-1K classification, $800{\times}1280$ for COCO object detection and instance segmentation, and $512 \times 2048$ for ADE20K semantic segmentation. The results are evaluated on NVIDIA 2080Ti GPUs. ('OOM' denotes out of GPU memory.)

| Model | Param (M) | Inference Time (ms) $224{\times}224$ | $800{\times}1280$ | $512{\times}2048$ |
|---|---|---|---|---|
| Low-resolution Baseline | | | | |
| ViT-D216-P16 | 7.4 | 0.9 | 155 | 150 |
| ViT-D348-P16 | 18.5 | 1.4 | 167 | 162 |
| ViT-D432-P16 | 28.5 | 1.8 | 177 | 173 |
| ViT-D624-P16 | 57.9 | 2.9 | 206 | 203 |
| High-resolution Baseline | | | | |
| ViT-D216-P8 | 7.4 | 7.5 | OOM | OOM |
| ViT-D348-P8 | 18.5 | 9.5 | OOM | OOM |
| ViT-D432-P8 | 28.5 | 11.3 | OOM | OOM |
| ViT-D624-P8 | 57.9 | 15.9 | OOM | OOM |
| GPViT-L1 | 9.3 | 2.7 | 83 | 87 |
| GPViT-L2 | 23.8 | 5.1 | 132 | 137 |
| GPViT-L3 | 36.2 | 7.0 | 174 | 182 |
| GPViT-L4 | 75.4 | 11.1 | 281 | 290 |

## A  FURTHER ABLATION STUDIES

### A.1  STUDY ON RUNNING EFFICIENCY

In Table 10, we compare the inference speed of GPViT with ViT baselines. Specifically, for each variant of our GPViT, we compare it to ViT models with patch size 16 (low-resolution) and patch size 8 (high-resolution) while keeping the channel dimensions the same. We report inference time using three different input sizes, which correspond to the three typical input sizes used by ImageNet-1k classification, COCO object detection and instance segmentation, and ADE20K semantic segmentation. We draw three observations from the results:

- When using small-sized inputs, GPViT runs slower than the low-resolution ViT. Despite the efficient design of our GP Block and the use of local attention, the high-resolution design still incurs a significant cost for forward passes, slowing down inference speed.

- When the models are applied to downstream tasks where they take larger-sized inputs, all but the largest GPViT models are faster than their low-resolution ViT counterparts. For example, when the model channel number is 216, GPViT takes 83 ms to process an $800{\times}1280$ sized image, while ViT-D216-P16 takes 155 ms. In this case, the self-attention operations with quadratic complexity severely slow down the speed of the ViT even with low resolution features. On the other hand, the computations in GP Block and local attentions grow much less than self-attention when the input scales up.

- GPViT is faster than the high-resolution ViT baselines when using small inputs. In addition, high-resolution ViTs are not even able to process large-sized inputs: we got *Out of Memory*

errors when using a NVIDIA 2080Ti GPU with 11 GB memory. This highlights our technical contribution of efficiently processing high-resolution features with GPViT.

We further study how the computation cost for high-resolution features changes when the model size and input scale up by examining FLOP counts. The results are shown in Figure 5 where we compare GP Block with different group numbers to self-attention and local-attention operations: Self-attention and GP Block can both exchange global information between image features, but the computational cost of GP Block grows much slower than self-attention. Local attention operations have a similar level of efficiency to GP Block, but are unable to exchange global information because of their limited receptive field.

## B IMPLEMENTATION DETAILS

### B.1 MODEL DETAILS OF GPVIT

The model details of different GPViT variants are presented in Table 11. Different GPViT variants are main difference by their model width (channels) and share similar hyper-parameters in other architecture designs.

Table 11: GPViT model details for different variants.

| Variants | GPViT-L1 | GPViT-L2 | GPViT-L3 | GPViT-L4 |
|---|---|---|---|---|
| Patch Size | 8 | 8 | 8 | 8 |
| Channel Dimension | 216 | 348 | 432 | 624 |
| Number of Transformer Layers | 12 | 12 | 12 | 12 |
| LePE Strip Size | 2 | 2 | 2 | 2 |
| Attention Heads | 12 | 12 | 12 | 12 |
| FFN Expansion | 4 | 4 | 4 | 4 |
| GP Block Positions | {1, 4, 7, 10} | {1, 4, 7, 10} | {1, 4, 7, 10} | {1, 4, 7, 10} |
| GP Block Group Numbers | {64, 32, 32, 16} | {64, 32, 32, 16} | {64, 32, 32, 16} | {64, 32, 32, 16} |
| Feature Grouping Attention Heads | 6 | 6 | 6 | 6 |
| Feature Ungrouping Attention Heads | 6 | 6 | 6 | 6 |
| MLPMixer Patch Expansion | 0.5 | 0.5 | 0.5 | 0.5 |
| MLPMixer Channel Expansion | 4 | 4 | 4 | 4 |
| ImageNet Drop Path Rate | 0.2 | 0.2 | 0.3 | 0.3 |
| Parameters (M) | 9.3 | 23.6 | 36.2 | 75.4 |

### B.2 TRAINING RECIPE FOR IMAGENET

The ImageNet experiments are based on the MMClassification toolkit (Contributors, 2020a). The models are trained for 300 epochs with a batch size of 2048; the AdamW optimizer was used with a weight decay of 0.05 and a peak learning rate of 0.002. The cosine learning rate schedule is adopted. The gradient clip is set to 5.0 (we also tested 1.0 and found it worked well too); data augmentation strategies are from Liu et al. (2021) and include Mixup (Zhang et al., 2017), Cutmix (Yun et al., 2019), Random erasing (Zhong et al., 2020) and Rand augment (Cubuk et al., 2020).

### B.3 TRAINING RECIPE FOR COCO

The COCO experiments are based on the MMDetection toolkit (Chen et al., 2019). Following commonly used training settings, both Mask R-CNN and RetinaNet models are trained for 12 epochs (1×) and 36 epochs (3×). For the 3× schedule, we follow previous work (Liu et al., 2021; Ren et al., 2022b) to use multi-scale inputs during training. The AdamW optimizer was used with an initial learning rate of 0.0002 and weight decay of 0.05. We used ViTAdapter (Chen et al., 2022b) to generate multi-scale features and followed the default hyper-parameter settings in Chen et al. (2022b).

### B.4 TRAINING RECIPE FOR ADE20K

The ADE20K experiments are based on the MMSegmentation toolkit (Contributors, 2020b). Following commonly used training settings, both UperNet and SegFormer models are trained for 160000 iterations. The input images are cropped to $512 \times 512$ during training. The AdamW optimizer was used with an initial learning rate of 0.00006 and weight decay of 0.01. We did not use ViTAdapter (Chen et al., 2022b) for segmentation experiments.

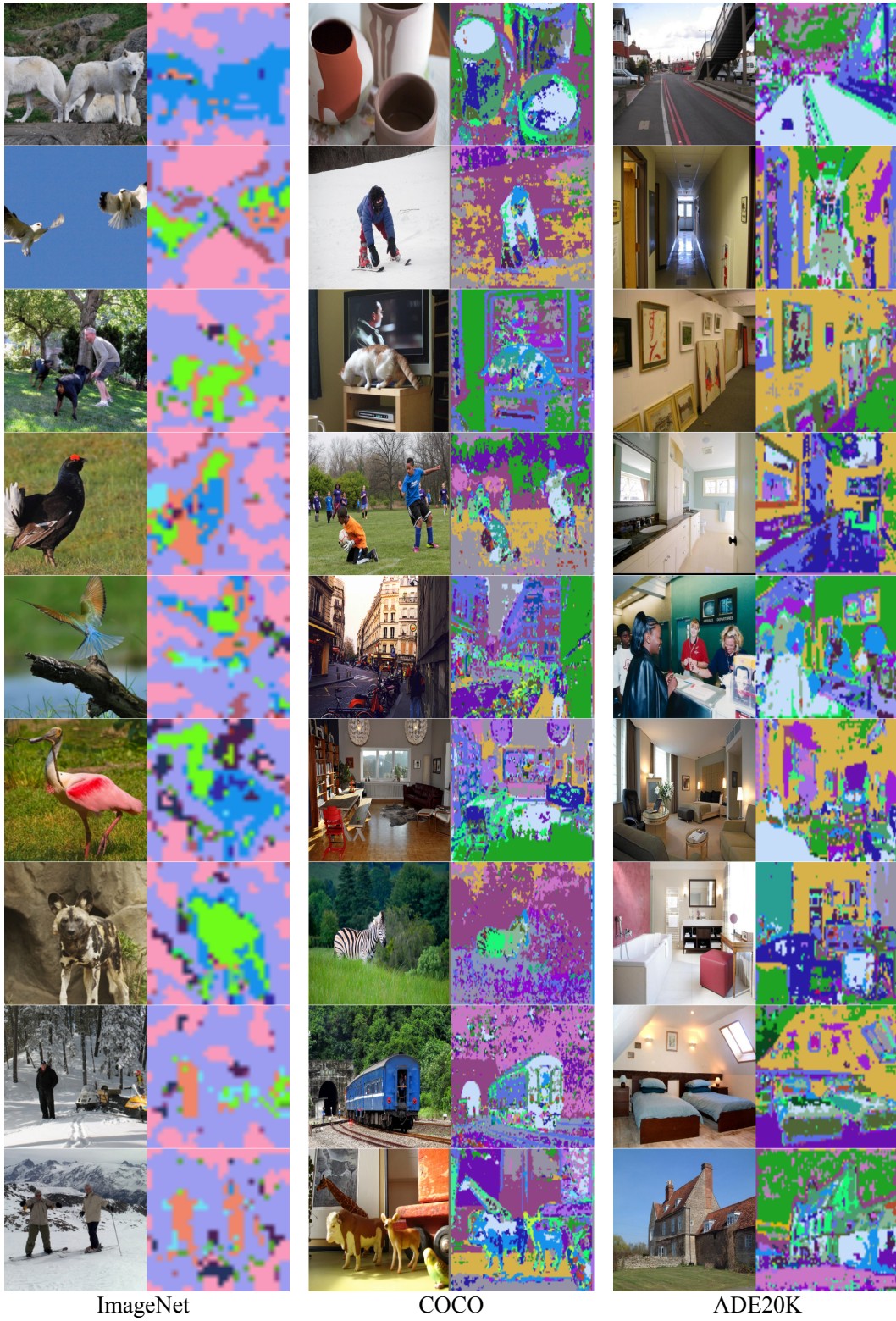

ImageNet        COCO        ADE20K

Figure 6: Feature grouping visualisation using model trained on ImageNet-1k, COCO and ADE20K.

## C  VISUALIZATIONS

In Figure 6, we visualise the feature grouping results using models trained on ImageNet, COCO and ADE20K. We observe that the feature grouping can separate a image's foreground and background in all three datasets. When the model receives fine-grained supervision like bounding boxes and semantic masks, the feature grouping can correspond to more details in the image.

# D COMPREHENSIVE COMPARISON

In Table 12 and Table 13, we provide a more comprehensive comparison between GPViT and other visual recognition models on ImageNet-1k classification and COCO Mask R-CNN object detection and instance segmentation.

Table 12: Comprehensive comparison between GPViT and the recent proposed models on ImageNet-1K.

| Model | Params (M) | FLOPs (G) | Top-1 Acc |
|---|---|---|---|
| **Hierarchical** | | | |
| PVT-T (Wang et al., 2021) | 13.2 | 1.9 | 75.1 |
| PVT-S (Wang et al., 2021) | 24.5 | 3.8 | 79.8 |
| PVT-L (Wang et al., 2021) | 64.1 | 9.8 | 81.7 |
| CvT-13 (Wu et al., 2021) | 20.0 | 4.5 | 81.6 |
| Focal-Tiny (Yang et al., 2021) | 29.1 | 4.9 | 82.2 |
| Focal-Base (Yang et al., 2021) | 89.8 | 16.0 | 83.8 |
| Swin-T (Liu et al., 2021) | 29 | 4.5 | 81.3 |
| Swin-S (Liu et al., 2021) | 50 | 8.7 | 83.0 |
| Swin-B (Liu et al., 2021) | 88 | 15.4 | 83.5 |
| ShiftViT-T (Wang et al., 2022a) | 29 | 4.5 | 81.3 |
| ShiftViT-B (Wang et al., 2022a) | 89 | 15.6 | 83.3 |
| CSwin-T (Dong et al., 2022) | 23 | 4.3 | 82.7 |
| CSwin-B (Dong et al., 2022) | 78 | 15.0 | 84.2 |
| RegionViT-Ti (Chen et al., 2022a) | 13.8 | 2.4 | 80.4 |
| RegionViT-S (Chen et al., 2022a) | 30.6 | 5.3 | 82.6 |
| RegionViT-B (Chen et al., 2022a) | 72.7 | 13.0 | 83.2 |
| MViTv2-T (Li et al., 2022b) | 24.0 | 4.7 | 82.3 |
| GC ViT-XXT (Hatamizadeh et al., 2022) | 12.0 | 2.1 | 79.8 |
| GC ViT-XT (Hatamizadeh et al., 2022) | 20.0 | 2.6 | 82.0 |
| GC ViT-T (Hatamizadeh et al., 2022) | 28.0 | 4.7 | 83.4 |
| GC ViT-S (Hatamizadeh et al., 2022) | 51 | 8.5 | 83.9 |
| GC ViT-B (Hatamizadeh et al., 2022) | 90 | 14.8 | 84.4 |
| DaViT-Tiny (Ding et al., 2022) | 28.3 | 4.5 | 82.8 |
| MPViT-S (Lee et al., 2022) | 22.8 | 4.7 | 83.0 |
| Shunted-T(Ren et al., 2022b) | 11.5 | 2.1 | 79.8 |
| Shunted-S(Ren et al., 2022b) | 22.4 | 4.9 | 82.9 |
| DW-T (Ren et al., 2022a) | 30.0 | 5.2 | 82.0 |
| DW-B (Ren et al., 2022a) | 91.0 | 17.0 | 83.8 |
| DAT-T (Xia et al., 2022) | 29.0 | 4.6 | 82.0 |
| DAT-B (Xia et al., 2022) | 88.0 | 15.8 | 84.0 |
| **Non-hierarchical** | | | |
| DeiT-S (Touvron et al., 2021a) | 22.1 | 4.6 | 79.9 |
| DeiT-B (Touvron et al., 2021a) | 86 | 16.8 | 81.8 |
| CaiT-XS24 (Touvron et al., 2021b) | 26.6 | 5.4 | 81.8 |
| CaiT-S48 (Touvron et al., 2021b) | 89.5 | 18.6 | 83.5 |
| LocalViT-S (Li et al., 2021b) | 22.4 | 4.6 | 80.8 |
| Visformer-S (Chen et al., 2021) | 40.2 | 4.9 | 82.3 |
| ConViT-S (d'Ascoli et al., 2021) | 27.0 | 5.4 | 81.3 |
| ConViT-B (d'Ascoli et al., 2021) | 86.0 | 17.0 | 82.4 |
| GPViT-L1 | 9.3 | 5.8 | 80.5 |
| GPViT-L2 | 23.8 | 15.0 | 83.4 |
| GPViT-L3 | 36.2 | 22.9 | 84.1 |
| GPViT-L4 | 75.4 | 48.2 | 84.3 |

Table 13: Comprehensive comparison of Mask R-CNN object detection and instance segmentation on MS COCO *mini-val* using 1× and 3× + MS schedule.

| Backbone | Params (M) | FLOPs (G) | 1× $AP^{bb}$ | 1× $AP^{mk}$ | 3× $AP^{bb}$ | 3× $AP^{mk}$ |
|---|---|---|---|---|---|---|
| **Hierarchical** | | | | | | |
| ResNet-50 (He et al., 2016) | 44 | 260 | 38.0 | 34.4 | 41.0 | 47.1 |
| ResNet101 (He et al., 2016) | 63 | 336 | 40.4 | 36.4 | 42.8 | 38.5 |
| ResNeXt101-32x4d (Xie et al., 2017) | 62 | 340 | 41.9 | 37.5 | 44.0 | 39.2 |
| RegionViT-S (Chen et al., 2022a) | 50.1 | 171 | 42.5 | 39.5 | 46.3 | 42.3 |
| RegionViT-B (Chen et al., 2022a) | 92 | 287 | 44.2 | 40.8 | 47.6 | 43.4 |
| PVT-Tiny (Wang et al., 2021) | 32.9 | - | 36.7 | 35.1 | 39.8 | 37.4 |
| PVT-Small (Wang et al., 2021) | 44.1 | - | 40.4 | 37.8 | 43.0 | 39.9 |
| PVT-Medium (Wang et al., 2021) | 63.9 | - | 42.0 | 39.0 | 44.2 | 40.5 |
| PVT-Large (Wang et al., 2021) | 81.0 | 364 | 42.9 | 39.5 | 44.5 | 40.7 |
| Swin-S (Liu et al., 2021) | 69 | 354 | 44.8 | 40.9 | 47.6 | 42.8 |
| Swin-B (Liu et al., 2021) | 107 | 496 | 45.5 | 41.3 | - | - |
| MViTv2-T (Li et al., 2022b) | 44 | 279 | - | - | 48.2 | 43.8 |
| MViTv2-S (Li et al., 2022b) | 54 | 326 | - | - | 49.9 | 45.1 |
| MViTv2-B (Li et al., 2022b) | 71 | 392 | - | - | 51.0 | 45.7 |
| DaViT-Small (Ding et al., 2022) | 69 | 351 | 47.7 | 42.9 | 49.5 | 44.3 |
| DaViT-Base (Ding et al., 2022) | 107 | 491 | 48.2 | 43.3 | 49.9 | 44.6 |
| DAT-T (Xia et al., 2022) | 48 | 272 | 44.4 | 40.4 | 47.1 | 42.4 |
| DAT-S (Xia et al., 2022) | 69 | 387 | 47.1 | 42.5 | 49.0 | 44.0 |
| DW-B (Ren et al., 2022a) | 111 | 505 | - | - | 49.2 | 44.0 |
| CMT-S (Guo et al., 2022) | 44 | 231 | 44.6 | 40.7 | - | - |
| CSwin-S (Dong et al., 2022) | 54 | 342 | 47.9 | 43.2 | 50.0 | 44.5 |
| CSwin-B (Dong et al., 2022) | 97 | 526 | 48.7 | 43.9 | 50.8 | 44.9 |
| MPViT-B (Lee et al., 2022) | 28 | 216 | - | - | 44.8 | 41.0 |
| MPViT-B (Lee et al., 2022) | 30 | 231 | - | - | 46.6 | 46.1 |
| MPViT-B (Lee et al., 2022) | 43 | 268 | - | - | 48.4 | 47.6 |
| MPViT-B (Lee et al., 2022) | 95 | 503 | - | - | 49.5 | 44.5 |
| Shunted-S(Ren et al., 2022b) | 42 | - | 47.1 | 52.1 | 49.1 | 43.9 |
| Shunted-B(Ren et al., 2022b) | 59 | - | 48.0 | 43.2 | 50.1 | 45.2 |
| GC ViT-T (Hatamizadeh et al., 2022) | 47 | 263 | - | - | 46.5 | 41.8 |
| **Non-hierarchical** | | | | | | |
| ViT-Adapter-T (Chen et al., 2022b) | 28 | - | 41.1 | 37.5 | 46.0 | 41.0 |
| ViT-Adapter-S (Chen et al., 2022b) | 47 | - | 44.7 | 39.9 | 48.2 | 42.8 |
| ViT-Adapter-B (Chen et al., 2022b) | 102 | - | 47.0 | 41.8 | 49.6 | 43.6 |
| GPViT-L1 | 33 | 457 | 48.1 | 42.7 | 50.2 | 44.3 |
| GPViT-L2 | 50 | 690 | 49.9 | 43.9 | 51.4 | 45.1 |
| GPViT-L3 | 64 | 884 | 50.4 | 44.4 | 51.6 | 45.2 |
| GPViT-L4 | 109 | 1489 | 51.0 | 45.0 | 52.1 | 45.7 |

