# OpenReview forum: "GPViT: A High Resolution Non-Hierarchical Vision Transformer with Group Propagation"
_ICLR.cc/2023/Conference — ICLR 2023 notable top 25%_

### Official Review · Reviewer_vWHX · 2022-10-21

**Confidence:** 3
**Correctness:** 3
**Technical Novelty And Significance:** 3
**Empirical Novelty And Significance:** 3
**Recommendation:** 8

**Clarity, Quality, Novelty And Reproducibility:**

See my comments above. Overall the paper is well written. it has a few items that need clarification but i do not think these constitute ground for rejection. The model is described in detail. It is not clear to me whether source code will be provided. A clarification from the authors is needed on this. The results are good and I think the ICLR audience would be interested in the paper.

**Strength And Weaknesses:**

Strengths:

S1: tested on a variety of tasks

S2: a good discussion on past transformer work, and the paper's place within this past work is discussed

S3: thorough testing and a thorough ablation study is performed. The authors show that their model achieves good accuracy. For imagenet however it is not always clear to me that there is an improvement in terms of the number of parameters used. Discuss this in more detail in the results



Weaknesses:

W1: unless I missed it, will source code be provided. I did not notice a discussion on whether source code will be provided to make it easy for someone to replicate these results.

W2: section 2: clarify what you mean by "inductive biases".

W3: Eq 2: where is the concat operation shown in Figure 2?

**Summary Of The Paper:**

The authors present GPViT, a non-hierarchival transformer model for visual recognition. The authors argue that by not using a hierarchical approach, and by relying only on high resolution features they achieve better recognition when fine grained details are needed for recognition. The authors introduce the Group Propagation Block to exchange global information tokens which achieves runtime efficiencies under these scenarios. They evaluate their approach on image classification. semantic segmentation object detection and instance segmentation.

**Summary Of The Review:**

As I indicated above, the results are good and I think the ICLR audience would be interested in the paper. The effort made to decrease the number of parameters should be appreciated by the ICLR audience.

---

> ### Author Response · Authors · 2022-11-18
> **Reply for Reviewer vWHX**
>
> We thank the reviewer for their thoughtful comments and suggestions. We address their questions below.
>
> **Q**: *“For imagenet however it is not always clear to me that there is an improvement in terms of the number of parameters used. Discuss this in more detail in the results”*
>
> **A**: Our models do scale well on ImageNet. For example, GPViT-L2 is 2.9% more accurate than GPViT-L1; GPViT-L3 is 0.7% more accurate than GPViT-L2, and the newest GPViT-L4 is 0.2% more accurate than GPViT-L3. These results are consistently better than other approaches with similar parameter totals. While the improvements on ImageNet are fairly modest, large gains are seen when we apply GPViTs to downstream tasks with high-resolution inputs. For example, on ADE20K, GPViT-L3 improves on GPViT-L2 by 1.5 mIoU. Please see more details on our new model GPViT-L4 in the revised paper.
>
> ---
>
> **Q**: *“Unless I missed it, will source code be provided. I did not notice a discussion on whether source code will be provided to make it easy for someone to replicate these results.”*
>
> **A**: Thank you for your interest in our code. We will open-source our code base and pre-trained model weights upon acceptance, as stated in our Reproducibility Statement (Page 9 in the paper).
>
> ---
>
> **Q**: *“Section 2: clarify what you mean by "inductive biases".”*
>
> **A**: Thank you for the suggestion. We have updated the paper in both Section 1 and 2 to clarify what we meant by CNN “inductive biases”, which are the properties of translation invariance, scale-invariance, and feature locality.
>
> ---
>
> **Q**: *“Eq 2: where is the concat operation shown in Figure 2?”*
>
> **A**: We have added colored indicators in Section 3 to connect the method explanations to Figure 2 following the reviewer’s question.  We hope those indicators can help you and other readers to digest our approach better.

---

### Official Review · Reviewer_oVeK · 2022-10-25

**Confidence:** 3
**Correctness:** 3
**Technical Novelty And Significance:** 2
**Empirical Novelty And Significance:** 2
**Recommendation:** 5

**Clarity, Quality, Novelty And Reproducibility:**

Overall the paper is pretty clear and easy to follow. Especially the overview figure of the architecture clearly explains the main building blocks. Even though I am not aware of approaches that follow the exact same approach, there are quite some very related existing methods that are not discussed here and I think this limits the novelty. Nevertheless a deeper view into this architecture could be interesting, but sadly the results here are more focused on showing "state of the art" performance. As such we also don't gain a lot of novel insights. Given the authors state that code will be released upon acceptance, I assume the results should be reproducible. Furthermore, the results are based on several MM toolkits, potentially further improving the reproducibility.

**Strength And Weaknesses:**

Strengths:
- While the idea to use such groups to reduce the complexity of attention is not really very novel, the application in this specific architecture is very interesting and show some promising results. Furthermore, a lot of the design choices seems rather general and similar systems could be applicable to other modalities such as point clouds, or potentially even multi-modal data.
- The overall direction of not using intermediate downsampling is interesting and worth investigating.
- The paper is easy to read and quite straightforward to follow.

Weaknesses:
- I think there is quite some rather related work that is not discussed. For example the Perceiver models by DeepMind have a different focus, but they are pretty similar in the underlying idea. Mask2Former also has a similar transformer part, albeit it works on top of a backbone to extract basic features and similar ideas where also explored in "Generative Adversarial Transformers". These are just a few I can recall off the top of my head. I wouldn't be surprised if there are a lot of other similar approaches. I think it is crucial that such related work is discussed with more care. Given that this is one main part of the contributions, the paper loses quite a bit of novelty in my mind. (And just because this uses an MLPMixer instead of vanilla attention does not make the model inherently different!)
- I'm quite sad to see yet another paper that simply claims a method achieves state of the art results on some task, when it's obviously not true. Simply looking at other approaches, it becomes clear that the numbers are far from state of the art. I know the focus here is likely on models with similar FLOP counts, but in most of the sentences where claims about being state of the art are made, this fact is simply omitted. It would be very important to clarify this! And apart from that, it would also be interesting to see how well this type of architecture generalizes to bigger versions of a model with similar FLOP counts than the actual state of the art. This should clearly be clarified more!
- Just looking at parameter counts and FLOP counts is not really giving a clear picture, but the actual throughput of a model is also important. (Have a look at the paper called "The Efficiency Misnomer".) Considering this architecture, I wouldn't be surprised if such a comparison would actually be favorable for the model, but I think it's sad that we don't see such numbers.
- I think the idea of the approach is interesting and it would be great to see a bit more experiments with respect to the design choices that went into the different building blocks. E.g. how many groups should be used, or why is the MLPMixer used instead of normal self-attention? Indeed some of these things are discussed in the appendix, but I think they should be featured more prominently in the main paper in order to highlight the important aspects of this architecture design.

**Summary Of The Paper:**

The paper presents a new transformer-based architecture that enables the extraction of high resolution feature maps from an images, while avoiding hierarchical downsampling in intermediate layers. To deal with the quadratic complexity of normal attention layers, a set of latent group representations is learned that collects information from the high resolution tokens. The latent group representations then mix their information based on an MLPMixer block and in turn the image patch tokens can cross attend to the update group features. Furthermore local attention is used for self-attention of the patch tokens. Using these principle, the complexity is linear in the number of groups and in the number of tokens, allowing a higher number of used input tokens. The resulting architecture is evaluated on several tasks (classification, detection, segmentation) and achieves good results compared to other methods with similar FLOP counts.

**Summary Of The Review:**

To me the core approach is interesting, but I'm not very convinced by the FLOP constrained evaluations. Furthermore some important related work is missing. Together these things limit the novelty and contributions quite a bit in my opinion and as such I am leaning more towards rejecting the paper. But I'm interested to see the other reviews and the rebuttal and would be willing to change my mind here.

---

> ### Author Response · Authors · 2022-11-18
> **Reply for Reviewer oVeK (Part 1)**
>
> We thank the reviewer for their thoughtful comments and suggestions. We hope to address their concerns below.
>
>
> **Q**: *“I think there is quite some rather related work that is not discussed, For example the Perceiver…. Mask2Former…”Generative Adversarial Transformers”…I think it is crucial that such related work is discussed with more care. Given that this is one main part of the contributions, the paper loses quite a bit of novelty in my mind.”*
>
> **A**: Thank you for pointing out this related work. We have included a discussion on these papers in the related work section of our revised paper. We respectfully disagree with the reviewer that this affects the novelty of our work. While e.g. “Mask2Former”[1] and “Perceiver”[2] do use transformer decoder layers with cross-attention between visual tokens and learnable tokens, as featured in our work, we note three fundamental differences: (i) each of our GP blocks operates as an `encoder-decoder' architecture with two rounds of cross-attention between visual tokens and group tokens: the first round groups the visual tokens for group propagation, and the second round ungroups the updated grouped features back into visual tokens; (ii) The underlying functionality is different: GP blocks facilitate more efficient global information propagation throughout the ViT, while previous work applies the decoder to obtain the final results for inference (e.g bounding boxes[3], or masks in [2]; (iii) The GP block is a general module that can be inserted into any layer of the ViT, while previous work utilizes the decoder only in the end of the network.
>
> Finally, we would like to emphasize again that our goal is to enable a general high-resolution non-hierarchical ViT backbone, instead of tackling a particular downstream application like Mask2Former.
>
> [1] Jaegle, Andrew, et al. "Perceiver: General perception with iterative attention." International conference on machine learning. PMLR, 2021.
>
> [2] Cheng, Bowen, et al. "Masked-attention mask transformer for universal image segmentation." Proceedings of the IEEE/CVF Conference on Computer Vision and Pattern Recognition. 2022.
>
> [3] Carion, Nicolas, et al. "End-to-end object detection with transformers." European conference on computer vision. Springer, Cham, 2020.
>
> ---
>
> **Q**: “I'm quite sad to see yet another paper that simply claims a method achieves state of the art results on some task, when it's obviously not true.”
>
> **A**: We thank the reviewer for the feedback. We agree this claim can cause confusion in the context of our paper. We have removed the state-of-the-art claims and replaced them with more precise expressions.
>
> ---
>
> **Q**: *“It would also be interesting to see how well this type of architecture generalizes to bigger versions of a model with similar FLOP counts than the actual state of the art.”*
>
> **A**: Given resource and time constraints, we are unable to scale up our model to sizes like that of ViT-E to compare GPViT with the true state-of-the-art. That said, we have added a larger model “GPViT-L4” to the updated paper, which is acquired by further increasing the channel number to 624. GPViT-L4 has 75.4M parameters, which is two times larger than GPViT-L3 (36.2M parameters). GPViT-L4 improves accuracy by 0.2% on the ImageNet-1k classification task compared to GPViT-L3. We achieve much more significant gains when applying GPViT-L4 to downstream tasks with high resolution inputs. On ADE20k semantic segmentation, the UperNet model equipped with GPViT-L4 achieve 52.5 mIoU, compared to 51.7 mIoU with  GPViT-L3. On COCO object detection and instance detection, GPViT-L4 Mask R-CNN achieves 51.0 AP$^{bb}$ and 45.0 AP$^{mk}$, improving over GPViT-L3’s 50.4 AP$^{bb}$ and 44.4 AP$^{mk}$. Furthermore, GPViT-L4 outperforms other models with similar parameter totals. This shows that our approach brings consistent improvements even as we scale up. Please see our revised paper for more details.

---

> ### Author Response · Authors · 2022-11-18
> **Reply for Reviewer oVeK (Part 2)**
>
> **Q**: *“Just looking at parameter counts and FLOP counts is not really giving a clear picture, but the actual throughput of a model is also important.”*
>
> **A**: We thank the reviewer for their suggestion. In the revised paper Appendix A1, we compare the inference speed of GPViT with high-resolution and low-resolution ViT baselines for different-sized inputs. Specifically, for each variant of our GPViT, we compare it to the ViT models with patch size 16 (low-resolution) and patch size 8 (high-resolution) while keeping the channel dimensions the same. We report inference time using three different input image sizes, which correspond to the three typical image input sizes used by ImageNet-1k classification, COCO object detection and instance segmentation, and ADE20K semantic segmentation. We draw three observations from the results:
> * When using small-sized inputs, GPViT is slower than the low-resolution ViT. The reason has been discussed in Section 3.2: Although our GP Block and the usage of local attentions can efficiently propagate global and local information, because of our high-resolution design, there is still considerable computation cost in the feed-forward network part, which slows down the inference speed;
> * When the models are applied to downstream tasks where they take larger-sized inputs, GPViT can become even faster than the low-resolution ViT. For example, when the model channel number is 216, GPViT takes 83 ms to process an 800×1280 sized image, while ViT-D216-P16 takes 155 ms to do that. In this case, the self-attention operations with quadratic complexity severely slow down the speed of ViT even if the features are in low resolution. On the other hand, the computations in our GP Block and local attention operations grow much less than self-attention when the input scales up, making GPViT becomes faster than the low-resolution ViT.
> * GPViT is faster than the high-resolution ViT baselines when using small inputs. In addition, high-resolution ViTs are not even able to process large-sized inputs: we got *Out of Memory* errors when using a NVIDIA 2080Ti GPU with 11 GB memory. This highlights our technical contribution of efficiently processing high-resolution features with GPViT.
>
> ---
> **Q**: *“I think the idea of the approach is interesting and it would be great to see a bit more experiments with respect to the design choices that went into the different building blocks. E.g. how many groups should be used, or why is the MLPMixer used instead of normal self-attention? Indeed some of these things are discussed in the appendix, but I think they should be featured more prominently in the main paper in order to highlight the important aspects of this architecture design.”*
>
> **A**: We thank the reviewer for their suggestion. We have moved the key ablation studies in the Appendix to the main paper. In Table 8, we show the performance of using different group numbers. In Table 9, we illustrate why we use MLPMixer: it can achieve similar accuracy to a  self-attention block, but use fewer FLOPs.

---

> ### Author Response · Authors · 2022-12-11
> **feedback**
>
> Dear reviewer,
>
> Since the meta-review is due soon. We are wondering if you can provide some feedback on our rebuttal, so that it can help the AC to write the meta-review.
>
> Thank you!
>
> Authors

---

### Official Review · Reviewer_6Qyc · 2022-10-29

**Confidence:** 4
**Clarity, Quality, Novelty And Reproducibility:** See  above.
**Correctness:** 4
**Technical Novelty And Significance:** 4
**Empirical Novelty And Significance:** 4
**Recommendation:** 10

**Strength And Weaknesses:**

Strengths:
1. The paper is clearly written. The method is explained well and carefully evaluated.
2. The idea of group propagation is intelligent and is effective at reducing computational cost while keeping the architecture simple.
3. The ablation studies and especially the explanation of how the architecture was built are informative and help explain the contribution of different components of the architecture.
4. The method obtains SOTA performance across tasks, with impressive gains when compared to prior methods with similar number of parameters or FLOPs.

Questions:
How would you scale up GPViT beyond L3 to obtain better performance? Are there marginal gains to be had on current datasets by further scale up?


**Summary Of The Paper:**

This paper proposes a non-hierarchical transformer model for visual recognition tasks (detection, segmentation). Unlike recently proposed hierarchical methods like Swin Transformer that use an hierarchical transformer architecture, exchanging global information between features is computationally expensive for non-hierarchical transformers. To deal with this challenge, the paper proposes an efficient  Group Propagation Block (GP Block) to exchange global information between high-resolution features. In a GP block, grouped features formed by learnable group tokens and then global information is exchanged between grouped features. Finally, global information in updated grouped features is returned to the image features through the transformer decoder. GPVIT is evaluated on  image classification, semantic segmentation, object detection, and instance segmentation and obtains state-of-the-art performance. Keeping parameters or FLOPs constant, GP-ViT shows improved performance over previous methods in all cases.


**Summary Of The Review:**

The paper proposes a novel, yet uncomplicated, architecture for non-hierarchical transformers for visual recognition tasks and obtains SOTA performance on these tasks.

---

> ### Author Response · Authors · 2022-11-18
> **Reply for Reviewer 6Qyc**
>
> **Q**: *“How would you scale up GPViT beyond L3 to obtain better performance? Are there marginal gains to be had on current datasets by further scale up?”*
>
> **A**: We thank the reviewer for their very positive review and valuable suggestion. To address the question on scaling, we have added a larger model “GPViT-L4” to the updated paper, which is acquired by further increasing the channel size to 624. GPViT-L4 has 75.4M parameters, which is two times larger than GPViT-L3 (36.2M parameters). GPViT-L4 improves accuracy by 0.2% on the ImageNet-1k classification task compared to GPViT-L3. We achieve much more significant gains when applying GPViT-L4 to downstream tasks with high-resolution inputs. On ADE20k semantic segmentation, the UperNet model equipped with GPViT-L4 achieved 52.5 mIoU, compared to 51.7 mIoU with  GPViT-L3. On COCO object detection and instance detection, GPViT-L4 Mask R-CNN achieves 51.0 AP$^{bb}$ and 45.0 AP$^{mk}$, improving over GPViT-L3’s 50.4 AP$^{bb}$ and 44.4 AP$^{mk}$. Furthermore, GPViT-L4 outperforms other models with similar parameter totals. This shows that our approach brings consistent improvements even as we scale up. Please see our revised paper for more details.

---

> > ### Comment · Reviewer_6Qyc · 2022-12-11
> > **Rebuttal response**
> >
> > Thank you for incorporating this experiment in the rebuttal. This is a good paper and I am happy to recommend acceptance.

---

### Author Response · Authors · 2022-11-18
**General reply to all reviewers:**

Dear reviewers,

We are very appreciative of your detailed comments and helpful suggestions. We have uploaded a revised version of the paper in light of your comments, where changes have been highlighted with red text.

Specifically, we have made the following updates in our revised paper:
1. We have added experimental results for GPViT-L4, a model with 75.4M parameters, over twice the size of GPViT-L3 (36.2M parameters) to answer Reviewer 6Qyc and Reviewer oVeK’s question on scaling, and we find our method continues to improve performance for the larger model.
2. We have added running speed results comparing GPViT with high-resolution and low-resolution ViT baselines following Reviewer oVeK’s suggestion.
3. We have expanded the section on related work following Reviewer oVeK’s comments.
4. We have replaced references to “state-of-the-art” with more precise expressions following Reviewer oVeK’s comments.
5. We move some ablation studies from the appendix into the main paper following Reviewer oVeK’s suggestion.
6. We clarify what we mean by “inductive biases” in the introduction and related work sections following Reviewer vWHX’s suggestion.

---

### Decision · Program_Chairs · 2023-01-20

**Decision:**

Accept: notable-top-25%

**Justification For Why Not Higher Score:**

The implementation of the proposed group-based self-attention is interesting but not groundbreaking.

**Justification For Why Not Lower Score:**

Nice implementation and comprehensive experiments would definitely attract the major audience in computer vision.

**Metareview: Summary, Strengths And Weaknesses:**

This paper presents a novel implementation of a group-based transformation of high-resolution features for general vision transformers. Experiments are comprehensive and the analysis is well-done.

Common Strengths:
1. Well-written.
2. A good implementation of using grouped features to save computation of self-attention.
3. Experiments are comprehensive, and of general interest to the community.

Weakness:
1. Missing important related works.
2. Missing some ablations.

**Note From Pc:**

if the above contains the word "oral" or "spotlight" please see: "oral" presentation means -> notable-top-5% and "spotlight" means -> notable-top-25%. As stated in our emails, we are disassociating presentation type from AC recommendations